# Consent is a confounding factor in a prospective observational study of critically ill elderly patients

Hans Flaatten[1]*, Bertrand Guidet[2], Christian Jung[3], Ariane Boumendil[4], Susannah Leaver[5], Wojciech Szczeklik[6], Antonio Artigas[7], Finn Andersen[8], Rui Moreno[9], Sten Walther[10], Sandra Oeyen[11], Joerg C. Schefold[12], Brian Marsh[13], Michael Joannidis[14], Muhammed Elhadi[15], Yuriy Nalapko[16], Jesper Fjølner[17], Dylan W. de Lange[18]

1 Department of Anaesthesia and Intensive Care, Haukeland University Hospital, Bergen, Norway, 2 Sorbonne Université, INSERM, Institut Pierre Louis d'Epidémiologie et de Santé Publique, Saint Antoine Hospital, AP-HP, Hôpital Saint-Antoine, Service de Animation, F75012, Paris, France, 3 Division of Cardiology, Pulmonology and Vascular Medicine, University Hospital Düsseldorf, Heinrich-Heine-University, Düsseldorf, Germany, 4 AP-HP, Hôpital Saint-Antoine, service de réanimation, F75012, Paris, France, 5 Research Lead Critical Care Directorate St George's Hospital, London, United Kingdom, 6 Intensive Care and Perioperative Medicine Division, Jagiellonian University Medical College, Kraków, Poland, 7 Intensive care Department, CIBER Enfermedades Respiratorias, Corporacion Sanitaria Universitaria Parc Tauli, Autonomous University of Barcelona, Sabadell, Spain, 8 Department of Anaesthesia and Intensive Care, NTNU, Dep of Circulation and Medical Imaging, Ålesund Hospital, Trondheim, Ålesund, Norway, 9 Faculdade de Ciências Médicas de Lisboa (Nova Médical School), Unidade de Cuidados Intensivos Neurocríticos e Trauma, Hospital de São José, Centro Hospitalar Universitário de Lisboa Central, Lisbon, Portugal, 10 Linkoping University Hospital, Linkoping, Sweden, 11 Department of Intensive Care 1K12IC Ghent University Hospital, Ghent, Belgium, 12 Department of Intensive Care Medicine, Inselspital, Universitätsspital, University of Bern, Bern, Switzerland, 13 Mater Misericordiae University Hospital, Dublin, Ireland, 14 Division of Intensive Care and Emergency Medicine, Department of Internal Medicine, Medical University Innsbruck, Innsbruck, Austria, 15 Faculty of Medicine, University of Tripili, Tripoli, Libya, 16 European Wellness International, ICU, Luhansk, Ukraine, 17 Department of Anaesthesia and Intensive Care, Viborg Regional Hospital, Viborg, Denmark, 18 Department of Intensive Care Medicine, University Medical Center, University Utrecht, Utrecht, The Netherlands

055 These authors contributed equally to this work.

* Hans.flaatten@uib.no

**Data Availability Statement:** Our data contains indirect possibilities to identify some patients since we have age, gender, country ID and ICU ID in our registry. Hence we have removed all potentially ID

## Abstract

During analysis of a prospective multinational observation study of critically ill patients ≥80 years of age, the VIP2 study, we also studied the effects of differences in country consent for study inclusion. This is a post hoc analysis where the ICUs were analyzed according to requirement for study consent. Group A: ICUs in countries with no requirement for consent at admission but with deferred consent in survivors. Group B: ICUs where some form of active consent at admission was necessary either from the patient or surrogates. Patients' characteristics, the severity of disease and outcome variables were compared. Totally 3098 patients were included from 21 countries. The median age was 84 years (IQR 81–87). England was not included because of changing criteria for consent during the study period. Group A (7 countries, 1200 patients), and group B (15 countries, 1898 patients) were comparable with age and gender distribution. Cognition was better preserved prior to admission in group B. Group A suffered from more organ dysfunction at admission compared to group B with Sequential Organ Failure Assessment score median 8 and 6 respectively. ICU

to patients and attached a file with data we hope will serve the purpose of your request but not putting anonymity in danger. These data is attached as a supplementary x-l file.

**Funding:** This study was endorsed by the ESICM. Free support for running the electronic database and was granted from the dep. of Epidemiology, University of Aarhus, Denmark. Financial support for creation of the e-CRF and maintenance of the database was possible from a grant (open project support) by Western Health region in Norway) 2018 project F-11487, who also funded the participating Norwegian ICUs. DRC Ile de France and URC Est helped conducting VIP2 in France. The VIP2-study collaborators are listed in the ESM 1.

**Competing interests:** Joerg C. Schefold declares that the Dept. of Intensive Care Medicine Bern has/ had research and/or development/consulting contracts with (full disclosure): Orion Corporation, Abbott Nutrition International, B. Braun Medical AG, CSEM SA, Edwards Lifesciences Services GmbH/SA, Kenta Biotech Ltd, Maquet Critical Care AB, Omnicare Clinical Research AG, and Nestlé. Educational grants were received from Fresenius Kabi; GSK; MSD; Lilly; Baxter; Astellas; AstraZeneca; B. Braun Medical AG, CSL Behring, Maquet, Novartis, Covidien, Nycomed, Pierre Fabre Pharma (Roba Pharma); Pfizer, Orion Pharma. The money went into departmental funds. No personal financial gain applies. This does not alter our adherence to PLOS ONE policies on sharing data and materials All other authors do not have any conflict of interest to declare related to this manuscript.

survival was lower in group A, 66.2% compared to 78.4% in group B (p<0.001). We hence found profound effects on outcomes according to differences in obtaining consent for this study. It seems that the most severely ill elderly patients were less often recruited to the study in group B. Hence the outcome measured as survival was higher in this group. We therefore conclude that consent likely is an important confounding factor for outcome evaluation in international studies focusing on old patients.

## Introduction

In a recently published, prospective, observational study of the acutely admitted very old intensive care patients ($\geq$ 80 years old), the VIP2 study) conducted in 22 European countries, we found an overall 30-day survival of 61%. Factors predicting mortality were frailty at admission, ICU admission categories and degree of organ failure at admission [1]. During the study, we also noticed a considerable heterogeneity among European countries regarding the requirements for informed consent for this observational study. Some countries waived the need for informed consent while others demanded consent prior to patient inclusion.

Previously we have reported problems with obtaining consent for a similar study with waiting time before decision up to one year for some countries [2].

In observational clinical research, the patients are typically not subjected to an intervention that can alter the course of the disease or illness. Only patient data that is already collected in the daily clinical routine or collected for the purpose of that prospective observational research, are used. In general, patients should have the right to decide whether they want their data to enter a research project, also in non-intervention studies. However, informed consent from patients is not always possible, particularly in acute care settings. When a patient is severely ill and admitted to an intensive care unit (ICU), they very often lack decisional capacity. In such instances, the national or regional medical ethical committee might agree to use informed consent from surrogate decision-makers or waive the need for informed consent at the time of admission, so-called deferred consent, and inform patient survivors about the study and their right to withdraw their inclusion at a later stage. The introduction of the General Data Protection Act (GDPR) in the European Union (EU) May 2018 necessitated in many countries an additional (written) informed consent for sharing privacy sensitive patent data even when deferred consent was allowed by the ethical committees. We started recruiting countries and ICUs to the VIP2 study in spring 2018, just before to the implementation of the European General Data Protection Regulation (GDPR). This resulted that some countries waived the need for informed consent while others, especially those who completed the ethical procedures after implementation of the GDPR, required written informed consent from either the critically ill patient themselves, or, if not possible, from next-of-kin.

During the preparation and analysis of the main publication from the VIP2 study, we were concerned about any effect from these two different approaches regarding consent or no consent with potential bias on the recruitment to the study.

In this paper, our aim was to reveal any effect from consent on patient demographics, in particular about presence of organ failure at admission and later mortality of the enrolled patients.

## Materials and methods

The methods specific for this post-hoc study are described below, and the detailed overview of the whole study can be found in the recent publication from the VIP-2 study [1]. The main aim of the VIP2 study was to investigate the influence of four common geriatric syndromes;

frailty, cognitive decline, activity of daily life and comorbidity/polypharmacy on various out-come measures, mainly ICU resource use and mortality. The severity of the critical illness was assessed by using the sequential organ dysfunction score (SOFA) [3] at admission.

To analyze potential effects from different requirements for ethical approval, we divided the countries in two groups, those allowed to include patients without consent at ICU admission (group A) and countries that needed patient or legal proxy consent at admission, be it from the patient, caregivers or independent physicians (Group B). In group A most countries required deferred consent in hospital survivors. To illustrate the effects, we analyzed the presence of geriatric syndromes clinical frailty scale (CFS), Katz activity of daily life, Informant Questionnaire on Cognitive Decline in the Elderly (IQCODE) and SOFA score at admission. The reason for admission to the ICU was grouped using a description of the main direct cause of admission as used previously. For outcomes, we used ICU length of stay (LOS), use of intensive care procedures like mechanical ventilation (MV) and use of vasoactive drugs (VAD) for organ support, use of limitation of care and ICU survival.

## Statistical methods

Baseline characteristics of patients were analysed as frequencies and percentages for categorical variables and as medians and interquartile ranges for continuous variables. Comparisons between consent and no consent group were evaluated using the Wilcoxon test for continuous variables and the χ2 or Fisher exact test for categorical variables as appropriate. The crude overall survival up to 90 days after ICU admission was estimated by the Kaplan-Meier method and compared between groups using a log-rank test.

Incidence of organ support and treatment limitations were estimated using cumulative incidence analysis considering ICU death and ICU discharge as competing risks. Univariate comparisons were performed using Gray's test.

In order to adjust comparison of outcomes for patients' characteristics Cox model was used for survival and cause-specific Cox models were used for organ support and treatment limitations.

The following factors were used for adjustment: age, sex, SOFA, reason for ICU admission, frailty and habitat.

Adjusted survival curves were produced using an Inverse Probability Weighted (IPW) Kaplan-Meier estimation. Significance was tested using a Cox regression model weighted by the same weights (inverse probability-weighted Cox).

We then used propensity score (PS) weighting to control for imbalances on observed variables between groups for all outcomes.

The PS model included the same covariates namely: age, sex, SOFA, reason for ICU admission, frailty and habitat. Generalized boosted regression were used to estimate the propensity score and cases were then weighted to estimate the average effect of consent on the population.

Two sensitivity analysis were conducted, first including all patients from England in the consent group, second including all patients from England in the no-consent group.

All tests were two-sided. The type-I error rate was fixed at 0.05. Analyses were performed using the R statistical software version 3.4.2 (R Foundation for Statistical Computing, Vienna, Austria [https://www.R-project.org/]); IPW analysis were performed using the ipwpoint function of the IPW package. PS analysis were performed using the PS function of the Twang package and weighted analyses used the survey package.

## Ethical statement

All 22 participating countries obtained ethical consent for conducting this study, although as the study shows the consent to participate varies from country to country. Of note is that 15

countries required informed consent from patient or proxy while 7 countries accepted deferred consent, that is consent after the patient had recovered. In Norway where the principal investigator (HF) works, the reference from the National Regional Board in Helse Sør-Øst is: 2018/87 (www.etikkom.no).

The main study was registered at ClinicalTrials.gov (ID: NCT03370692).

## Results and discussion

The main study (VIP2) recruited 3920 patients with a mean age of 84 years (95% CI 84–85 years) from 242 ICUs in 22 countries. The median duration of patient recruitment in an ICU was 64 days with no differences in outcomes in the ICUs below or above the median duration of recruitment.

One country, England, initially required full consent but later accepted inclusion without consent for patients who died prior to consent. Since we have no information from individual ICUs the UK data is excluded from primary analysis leaving 3098 patients in this substudy. Hence, this country was excluded from the main analysis but included in a sensitivity analysis since they were the largest contributing country with 822 recruited patients.

Another country, the Netherlands, initially approved to include patients without consent, but after the introduction of GDPR, 5 hospitals changed this to informed consent. Hence this country appears with ICUs in both groups. The group A consisted of 7 countries with 1200 patients included and group B of 15 countries with 1898 patients. Table 1 shows the baseline demographic and clinical data in the two groups, and Table 2 shows the differences in the six SOFA sub-scores.

The groups were comparable at admission with regards to age and gender. SOFA score was higher in group A. Also, the cognitive decline score, IQCODE, was also increased, suggesting worse cognition in this group. With regards to outcomes, the groups differ. Patients in group A with higher degree of organ dysfunction were more often mechanically ventilated and given vasoactive drugs at day 1 and ICU survival was lower than in group B (66.2% vs 78.4%).

**Table 1. Demographic and clinical data in all patients.** Data given as median and IQR.

|  | Consent (Group B)[b] | No consent (group A)[a] |
|---|---|---|
| Patients | 1898 | 1200 |
| Countries | 15 | 7 |
| Age in years, median (IQR) | 84 (81–87) | 84 (81–86) |
| Gender (female) n (%) | 872 (45.9%) | 567 (47.2%) |
| SOFA, median (IQR) [1] | 6 (3–9) | 8 (5–10) |
| CFS, median (IQR) | 4 (1–9) | 4 (1–9) |
| IQCODE, median (IQR)[1] | 3.19 (3–3.62) | 3,31 (3.06–4.0) |
| Katz ADL, median (IQR)[1] | 6 (4–6) | 6 (3–6) |
| Mechanical ventilation n (%) [1] | 907 (47.7%) | 738 (61.6%) |
| Vasoactive drugs[1] | 993(52.4%) | 894 (74.7%) |
| Withdrawal | 231 (12.3%) | 122 (10.3%) |
| ICU survival | 1484 (78.4%)[1] | 787 (66.2%) |

[1] p<0.0001.

SOFA = Sequential Organ Failure Assessment; CFS = Clinical Frailty Scale; IQCODE = Informant Questionnaire on Cognitive Decline in the Elderly; ADL = Activity of Daily Life.

[a] Norway, Sweden, Poland, Greece, Libya, Austria and Netherland (partially)

[b] Denmark, Ireland, Wales, Germany, Belgium, France, Switzerland, Spain, Italy, Portugal, Ukraine, Russia, Turkey, Croatia, Netherlands (partially).

Table 2. SOFA sub-score in the two groups. Data given as median and IQR.

| SOFA sub-score | Consent (group B) | No consent (group A) |
|---|---|---|
| SOFA resp | 2 (1–3) | 2 (1–3) [1] |
| SOFA circ | 0 (0–2) | 1 (0–3) [1] |
| SOFA neuro | 0 (0–4) | 1 (0–4) [1] |
| SOFA liver | 0 (0–0) | 0 (0–0) |
| SOFA coag | 0 (0–0) | 0 (0–1) [1] |
| SOFA renal | 1 (0–2) | 1 (0–2) |

[1] p<0.0001.

There were differences in admission categories as can be seen from Table 3 with more single respiratory failure in group B while there was more combined respiratory and circulatory failure in group A. No clinical differences in end-of-life decisions between the groups were found.

In Table 4 the unadjusted and adjusted HR for outcomes in the consent and no consent group are given. Both models confirm significantly higher risk of death in the no-consent group. Also, the propensity score weighting yield the same results. The table also shows that common ICU procedures like mechanical ventilation and vasoactive drugs support were more frequently used in the non-consent group.

The adjusted cumulative survival curves between the two groups are illustrated in Fig 1 and clearly demonstrate the continuing difference up to 180 days.

In the sensitivity analysis, we included England sequentially to either group A or B. If included in the non-consent group the results were not altered, but if all were included in the consent group the differences were no longer significant (Table 1a and 1b in S1 Table). In this secondary analysis of data from the VIP2 study, we found large differences in the patient cohorts and hence outcomes when countries were compared with versus without required upfront consent for study inclusion. Despite similar age, they have more cognitive dysfunction and reduced activity of daily life. The most striking difference is the increased SOFA score indicating more severe organ dysfunctions in this group. We also found differences in the specter of admission groups where more patients with isolated respiratory failure was admitted in group B but more combined respiratory and circulatory failure in group A. The combined admission group is usually more critical ill and in need of ventilatory support and vasoactive drugs. As a consequence, also after correction for the known confounding factors, survival in the two groups is significantly different, with 66.2% survivors when no informed consent was requested necessary compared to 78.4% in the consent group, an absolute difference in mortality of 12.2%.

Table 3. Admission categories in the two groups (n and % within the group).

| Reason for admission | Consent (group B) | Non-consent (group A) |
|---|---|---|
| Respiratory failure | 520 (27.4%) | 252 (21%) |
| Circulatory failure | 256 (13.5%) | 179 (14.9%) |
| Combined Resp&Circ failure | 181 (9.5%) | 200 (16.7%) |
| Sepsis (Sepsis 3) | 270 (14.2%) | 188 (15.7%) |
| Trauma | 94 (5.1%) | 89 (7.5%) |
| Intoxication | 20 (1.1%) | 1 (0.1%) |
| Non-trauma CNS disease | 120 (6.3%) | 50 (4.2%) |
| Emergency surgery | 191 (10.1%) | 136 (11.3%) |

**Table 4. Results from the Cox analysis and propensity score weighting in the consent versus non-consent groups.**

| UNADJUSTED | | | |
|---|---|---|---|
| | HR consent vs no consent | 95% CI | P-value |
| Overall survival | 0.66 | 0.59–0.73 | <0.0001 |
| **ADJUSTED Cox [1]** | | | |
| | HR consent vs no consent | 95% CI | P-value |
| Overall survival | 0.82 | 0.74–0.91 | 0.0027 |
| **IPW weighted Cox model [1]** | | | |
| | HR consent vs no consent | 95% CI | P-value |
| Overall survival | 0.84 | 0.74–0.94 | 0.0023 |
| **Propensity score weighting [1]** | HR consent vs no consent | 95% CI | P-value |
| Overall survival | 0.81 | 0.72–0.90 | 0.0002 |
| Mechanical ventilation | 0.84 | 0.76–0.93 | 0.0004 |
| Vasoactive drugs | 0.71 | 0.65–0.79 | <0.0001 |
| Renal replacement | 0.89 | 0.71–1.10 | 0.257 |
| Limitation of care | 1.05 | 0.91–1.2 | 0.472 |

[1] including variables age, gender, habitat, frailty, reason for ICU admission and SOFA score.

One likely explanation for this finding is difficulties in recruiting the most unstable patients when informed consent was required at ICU admission. This seems logical for several reasons. When a critically ill patient is admitted to the ICU, the focus is on the acute treatment and stabilization of the disease process. Confronting patients or caregivers/family in this phase with information and explanation about a clinical study, although an observational study with no intervention, often has low priority. If directly asked, many patients are unconscious or with severely reduced mental capacity and caregivers are often too stressed to digest and understand information when asked for surrogate consent. Additionally, in many tertiary centers, family and caregivers may arrive in the ICU after the inclusion window, precluding participation in the study. Hence, sometimes the simplest way to respond is not to give consent. However, this

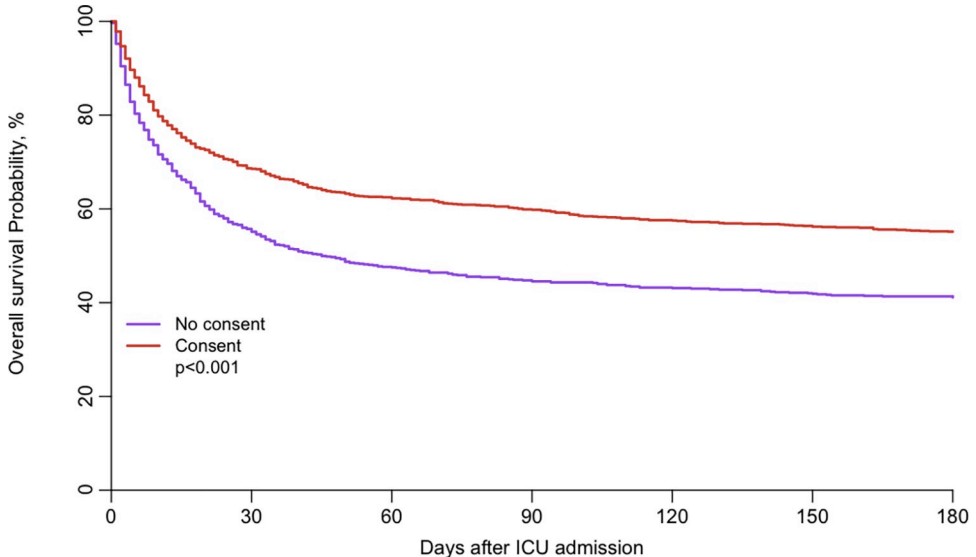

**Fig 1. Adjusted survival curve in the consent and no consent groups.** (Fig 1 OS-adjusted-consent.tiff).

does not mean that such an ICU has equally sick patients, but for the abovementioned reasons, they are just not recruited into the study, resulting in a selection bias of less severely ill patients.

As we have demonstrated, the way consent is handled seems to be an important and independent confounding factor with considerable implication for the composition of the recruited cohort and hence affecting the primary study endpoint: survival. If the most critically ill unstable patients more often are left out from inclusion in the study and further analysis this may give a false impression of a better outcome.

Consent has been the focus in several publications [4], but not discussed in clinical studies involving critically ill old patients. In a study about consent rates documented in clinical studies involving critically ill patients, the researchers found that, particularly in non-randomized studies, information on how consent was handled was lacking in most studies (81%). This was considered a potential source of bias and validity of the studies [5]. The use of surrogates for informed consent in patients on mechanical ventilation has recently been discussed and was associated with difficulties and was not always consistent with the patients' view [6]. In an interesting discussion about implication of the Food and Drug Administration guideline for informed consent, the authors conclude that inclusion without patient consent should be feasible with the current guideline when patients cannot consent [7].

In some countries, like in Norway, there is a special document dealing with inclusion of patients in studies concerning emergency medical conditions [7]. Research, even intervention trials, may be conducted if: 1. The patient is unable to consent (unconscious, unable to comprehend information). 2. Similar research cannot be done in non-emergency situations; and that the project has been approved from the independent local ethical committee. 3. Research should be beneficial for the patient or should be of potential benefit for others and impose minimal extra risks for included patients. 4. If the patient recovers, he/she must be given information about the study at that time and the option to consent or not (deferred consent). However, as our study reveals, in most European countries observational research cannot be performed today without written informed consent at admission. The fact that this is obstructing observational research has been discussed previously [8,9]

Our study was a purely observational study with no interventions outside standardizing information about geriatric syndromes. Although not potentially beneficial or detrimental for the individual patient, our study´s information may have huge implication for the group of very old patients who are acutely admitted to the ICU. If, as our results may indicate, survival in unselected patients is nearly halved within 30 days, we have the obligation to find ways to reduce unnecessary and futile therapy that is a huge burden on patients as well as caregivers, and to provide such care to those that most likely will profit from such treatment. Moreover, if the external validity of future observational studies is so low that we cannot trust such information, then our research becomes useless or even dangerous.

This study has its strength being a prospective study with a high number of patients from many different countries in Europe, and patients were followed for 30 days. However, a weakness is the post-hoc design of this sub-study, which was not planned for during the pre-study phase. Another weakness is that in some countries with no national guidelines for research Ethical boards, the rules for inclusion could vary from region to region, and this variation have not been possible to capture in this analysis.

## Conclusions

In this study we document the effects on how to receive informed consent in two patient groups within the same prospective study. If consent was necessary at admission (compared to deferred consent patients), patients were less severely ill and with a higher 30-day survival.

We hope that the EU will engage in the harmonization of inclusion rules in prospective observational studies where patients are unable to understand and/or make their own judgment regarding study participation. The unfortunate alternative would be significant biased data obtained from critically ill patients in Europe.

## Supporting information

**S1 Table. Table 1 Showing results from the Cox analysis and propensity score weighting in the consent versus non-consent groups when England was added to consent or non-consent group.**
(DOCX)

**S1 Dataset. A file with de-identified data performed in the analysis (minimal underlying dataset.xls).**
(XLS)

## Author Contributions

**Conceptualization:** Hans Flaatten, Bertrand Guidet, Dylan W. de Lange.

**Data curation:** Hans Flaatten, Christian Jung, Antonio Artigas, Finn Andersen, Jesper Fjølner.

**Formal analysis:** Hans Flaatten, Bertrand Guidet, Ariane Boumendil.

**Funding acquisition:** Hans Flaatten, Bertrand Guidet.

**Investigation:** Bertrand Guidet, Christian Jung, Jesper Fjølner.

**Methodology:** Hans Flaatten, Christian Jung, Ariane Boumendil, Jesper Fjølner, Dylan W. de Lange.

**Project administration:** Hans Flaatten, Bertrand Guidet, Christian Jung, Susannah Leaver, Wojciech Szczeklik, Finn Andersen, Rui Moreno, Sten Walther, Joerg C. Schefold, Brian Marsh, Michael Joannidis, Muhammed Elhadi, Yuriy Nalapko, Jesper Fjølner, Dylan W. de Lange.

**Resources:** Hans Flaatten, Bertrand Guidet.

**Software:** Ariane Boumendil.

**Supervision:** Hans Flaatten, Christian Jung, Wojciech Szczeklik, Antonio Artigas, Finn Andersen, Rui Moreno, Sten Walther, Sandra Oeyen, Joerg C. Schefold, Brian Marsh, Michael Joannidis, Muhammed Elhadi, Yuriy Nalapko, Jesper Fjølner.

**Validation:** Hans Flaatten, Bertrand Guidet, Susannah Leaver, Wojciech Szczeklik, Antonio Artigas, Rui Moreno, Dylan W. de Lange.

**Visualization:** Hans Flaatten.

**Writing – original draft:** Hans Flaatten.

**Writing – review & editing:** Bertrand Guidet, Christian Jung, Ariane Boumendil, Susannah Leaver, Wojciech Szczeklik, Antonio Artigas, Finn Andersen, Rui Moreno, Sten Walther, Sandra Oeyen, Joerg C. Schefold, Brian Marsh, Michael Joannidis, Muhammed Elhadi, Yuriy Nalapko, Jesper Fjølner, Dylan W. de Lange.

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
