## [Decision Letter · Decision Letter 0]

9 May 2022

PONE-D-22-08080Consent is a confounding factor in a prospective observational study of critical ill elderly patientsPLOS ONE

Dear Dr. Flaatten,

Thank you for submitting your manuscript to PLOS ONE. After careful consideration, we feel that it has merit but does not fully meet PLOS ONE’s publication criteria as it currently stands. Therefore, we invite you to submit a revised version of the manuscript that addresses the points raised during the review process.

 The paper that you submitted to PLOS ONE has been seen by three referees whose reports are listed below. Your paper addresses an intriguing question, but as you will see, the statistician referee #1 raised several important criticisms which make the paper unacceptable for publication in its present form. Despite the limtations of the study, I personally believe that the authors have the responsibility to clearly answer to the issue raised in this very important and strategic study. Please submit your revised manuscript by Jun 23 2022 11:59PM. If you will need more time than this to complete your revisions, please reply to this message or contact the journal office at plosone@plos.org. Please include the following items when submitting your revised manuscript:A rebuttal letter that responds to each point raised by the academic editor and reviewer(s). You should upload this letter as a separate file labeled 'Response to Reviewers'.A marked-up copy of your manuscript that highlights changes made to the original version. You should upload this as a separate file labeled 'Revised Manuscript with Track Changes'.An unmarked version of your revised paper without tracked changes. You should upload this as a separate file labeled 'Manuscript'.

We look forward to receiving your revised manuscript.

Kind regards,

Giuseppe Vittorio De Socio, MD, PhD

Academic Editor

PLOS ONE

Journal Requirements:

2. If materials, methods, and protocols are well established, authors may cite articles where those protocols are described in detail, but your submission should include sufficient information to be understood independent of these references (https://journals.plos.org/plosone/s/submission-guidelines#loc-materials-and-methods).

"Joerg C. Schefold declares that the Dept. of Intensive Care Medicine Bern has/ had research and/or development/consulting contracts with (full disclosure): Orion Corporation, Abbott Nutrition International, B. Braun Medical AG, CSEM SA, Edwards Lifesciences Services GmbH/SA, Kenta Biotech Ltd, Maquet Critical Care AB, Omnicare Clinical Research AG, and Nestlé. Educational grants were received from Fresenius Kabi; GSK; MSD; Lilly; Baxter; Astellas; AstraZeneca; B. Braun Medical AG, CSL Behring, Maquet, Novartis, Covidien, Nycomed, Pierre Fabre Pharma (Roba Pharma); Pfizer, Orion Pharma. The money went into departmental funds. No personal financial gain applies."

All other authors do not have any conflict of interest to declare related to this manuscript.

7. Your ethics statement should only appear in the Methods section of your manuscript. If your ethics statement is written in any section besides the Methods, please move it to the Methods section and delete it from any other section. Please ensure that your ethics statement is included in your manuscript, as the ethics statement entered into the online submission form will not be published alongside your manuscript.

Reviewers' comments:

Reviewer's Responses to Questions

**Comments to the Author**

1. Is the manuscript technically sound, and do the data support the conclusions?

Reviewer #1: No

Reviewer #2: Yes

Reviewer #3: Yes

2. Has the statistical analysis been performed appropriately and rigorously? 

Reviewer #1: No

Reviewer #2: Yes

Reviewer #3: Yes

3. Have the authors made all data underlying the findings in their manuscript fully available?

Reviewer #1: No

Reviewer #2: Yes

Reviewer #3: No

4. Is the manuscript presented in an intelligible fashion and written in standard English?

Reviewer #1: No

Reviewer #2: Yes

Reviewer #3: Yes

5. Review Comments to the Author

Reviewer #1: Current paper presented an observational study comparing ICU length of stay (LOS), use of intensive care procedures for organ support and 30-day survival. The paper is written in hard to read English with sentences possibly grammatically correct but should be revised by a native speaker for easy reading. In the next section I am primarily focusing on the statistical and data analysis aspect of the project.

1. The paper gives very little information about the observational data to be analyzed. Table 1 presents some data but that is only the tip of the iceberg. Please give detailed demographic and clinical data as well as their standard deviation (when applicable). For some variable mean and for some median is reported without reasoning. For observational data analysis detailed description of the data is must, so that a reader can understand what is being analyzed.

2. The statistical methods used for analysis are either cryptic or without sufficient description. It seems only two group comparisons are made without addressing inherent bias (typical in any observational data) in the data. A propensity score or similar methods should have been used instead.

3. How the figure 1 is produced is not clear. The paper also mentions “sensitivity analysis” and “regression analysis” in passing without going into detail about the results or how it is being carried out.

In short, the paper's statistical methods and its interpretation as well as presentation is below the publication standard. It is highly recommended to include a trained statistician to curate these areas.

Reviewer #2: This is and interesting and original paper on a very important topic: the consent to participate to clinical studies.

The Authors show that the lack of consent is related to scientific biases and to poor outcomes in the older sample of patients of this research.

The paper is well written and the English form is correct.

Data are well expressed and the results are exposed in a clear form. The discussion of results is very clear.

Reviewer #3: The authors should be congratulated because the issue they address in this paper is methodologically relevant. The manuscript deals with analysis of potential selection bias and, more than relevant for the VIP2 study, the issue is relevant in general for clinical research in the acute medicine setting.

I have some issues that can improve the quality of the manuscript and the knowledge on the issue:

- please describe cause of admission for the two groups (A vs B)

- please describe the components of the SOFA score in the two groups (A and B)

6. PLOS authors have the option to publish the peer review history of their article (what does this mean?). If published, this will include your full peer review and any attached files.

Reviewer #1: No

Reviewer #2: No

Reviewer #3: No

---

## [Author Response · Author response to Decision Letter 0]

24 Jun 2022

We have uploaded our response to the referees as well as the editor in word files.

---

## [Decision Letter · Decision Letter 1]

24 Aug 2022

PONE-D-22-08080R1Consent is a confounding factor in a prospective observational study of critically ill elderly patientsPLOS ONE

Dear Dr. Flaatten,

Thank you for submitting your manuscript to PLOS ONE. After careful consideration, we feel that it has merit but does not fully meet PLOS ONE’s publication criteria as it currently stands. Therefore, we invite you to submit a revised version of the manuscript that addresses the points raised during the review process.

We really appreciate the efforts made by the authors however, as You will see that the statistical Referee  also raised major criticisms and advised against the publication of this paper. Please respond to all the comments by Reviewer #4, with special attention to methodological points. If you think all objections raised by the Referee can be considered, we may be willing to reconsider your manuscript. Yet, bear in mind that your paper will be sent out for further scrutiny to referees.

We look forward to receiving your revised manuscript.

Kind regards,

Giuseppe Vittorio De Socio, MD, PhD

Academic Editor

PLOS ONE

Reviewers' comments:

Reviewer's Responses to Questions

**Comments to the Author**

1. If the authors have adequately addressed your comments raised in a previous round of review and you feel that this manuscript is now acceptable for publication, you may indicate that here to bypass the “Comments to the Author” section, enter your conflict of interest statement in the “Confidential to Editor” section, and submit your "Accept" recommendation.

Reviewer #2: All comments have been addressed

Reviewer #4: (No Response)

2. Is the manuscript technically sound, and do the data support the conclusions?

Reviewer #2: Yes

Reviewer #4: Partly

3. Has the statistical analysis been performed appropriately and rigorously? 

Reviewer #2: Yes

Reviewer #4: No

4. Have the authors made all data underlying the findings in their manuscript fully available?

Reviewer #2: Yes

Reviewer #4: No

5. Is the manuscript presented in an intelligible fashion and written in standard English?

Reviewer #2: Yes

Reviewer #4: Yes

6. Review Comments to the Author

Reviewer #2: The paper is well written, the results are clear, the discussion is good.

I was already convinced to accept the paper in the previous form.

Reviewer #4: This manuscript describes a secondary data analysis study for a prospective, observational study of the acutely admitted very old intensive care patients (the VIP2 study). The findings in the original VIP2 study (conducted in 22 European countries) was recently published. In this secondary data analysis study, the authors identified the methods for Information Consent as a major confounding factor in prospective observational studies of critically ill elderly patients. The finding is interesting and potentially useful for readers. The statistical analysis methods are relatively simple but reasonable.

However, I have some concerns:

Major concerns:

1. In table 1, some numbers seem problematic, for example, the last three row:

SOFA median (95% CI) 8 (8-9) 6 (6-7)

CFSa median (95% CI) 4 (4-5) 4 (4-5)

IQCODEb median (95% CI) 3.31 (3.31-3.38) 3.19 (3.19-3.25)

How could the lower bound of the 95% confidence interval (i.e. the 2.5% quantile) be the same as the sample median (50% quantile)? Although this is potentially possible under some rare scenarios, it is more likely due to some careless mistakes in data analysis. If the authors do think the current results are correct, please show the histogram of these variables, in the response letter, so that reviewers can confirm the results.

2. The references in the revised manuscript seem to be misplaced. For example, reference [3] in the main text seem to be the wrong reference (it should be the reference [4] in the reference list. In addition there 9 references in the reference list, only 8 were referred in the main text.

3. It was mentioned in the last sentence of the Result section that “In the regression analysis consent or no consent was an independent variable in addition to SOFA score and IQCODE.” However, the results of the regression analysis were not presented anywhere (as either table or figure) in the manuscript.

Minor concerns:

4. Please spell out which analysis were used to calculate the p value in Figure 1.

5. The first sentence of the Introduction, the parentheses do not match.

“In a recently published, prospective, observational study of the acutely admitted very old intensive care patients (≥ 80 years old), the VIP2 study) conducted in 22 European countries, we found an overall 30-day survival of 61%. Factors predicting mortality were frailty at admission, ICU admission categories and degree of organ failure at admission [1].”

7. PLOS authors have the option to publish the peer review history of their article (what does this mean?). If published, this will include your full peer review and any attached files.

Reviewer #2: **Yes: **Chiara Mussi

Reviewer #4: No

---

## [Author Response · Author response to Decision Letter 1]

15 Sep 2022

As stated in the letter there was an error from my side in files sent for revision 1. By a mistake the clean version was the original clean manuscript while the version with changes visible was the right one. In this submission the clean version is the correct clean version based on all the changes done in R1. Hence the comments from reviewer 4 in the original submission was already taken care of but as stated not seen in the clean version. This is now corrected so the both submitted manuscripts, clean and with visible correction, are now the same, and the comments from reviewer 4 hence are taken care of.

Again, I am sorry for this mistake from my side.

---

## [Decision Letter · Decision Letter 2]

6 Oct 2022

Consent is a confounding factor in a prospective observational study of critically ill elderly patients

PONE-D-22-08080R2

Dear Dr. Flaatten,

We’re pleased to inform you that your manuscript has been judged scientifically suitable for publication and will be formally accepted for publication once it meets all outstanding technical requirements.

Kind regards,

Giuseppe Vittorio De Socio, MD, PhD

Academic Editor

PLOS ONE

Additional Editor Comments (optional):

Reviewers' comments:

Reviewer's Responses to Questions

**Comments to the Author**

1. If the authors have adequately addressed your comments raised in a previous round of review and you feel that this manuscript is now acceptable for publication, you may indicate that here to bypass the “Comments to the Author” section, enter your conflict of interest statement in the “Confidential to Editor” section, and submit your "Accept" recommendation.

Reviewer #4: All comments have been addressed

2. Is the manuscript technically sound, and do the data support the conclusions?

Reviewer #4: (No Response)

3. Has the statistical analysis been performed appropriately and rigorously? 

Reviewer #4: (No Response)

4. Have the authors made all data underlying the findings in their manuscript fully available?

Reviewer #4: (No Response)

5. Is the manuscript presented in an intelligible fashion and written in standard English?

Reviewer #4: (No Response)

6. Review Comments to the Author

Reviewer #4: (No Response)

7. PLOS authors have the option to publish the peer review history of their article (what does this mean?). If published, this will include your full peer review and any attached files.

Reviewer #4: No

---

## [Editor Report · Acceptance letter]

12 Oct 2022

PONE-D-22-08080R2 

Consent is a confounding factor in a prospective observational study of critically ill elderly patients 

Dear Dr. Flaatten:

I'm pleased to inform you that your manuscript has been deemed suitable for publication in PLOS ONE. Congratulations! Your manuscript is now with our production department. 

Kind regards, 

on behalf of

Dr. Giuseppe Vittorio De Socio 

Academic Editor

PLOS ONE